# Genotype-Specific Antioxidant Responses and Assessment of Resistance Against *Sclerotinia sclerotiorum* Causing Sclerotinia Rot in Indian Mustard

**DOI:** 10.3390/pathogens9110892

**Published:** 2020-10-27

**Authors:** Manjeet Singh, Ram Avtar, Ajay Pal, Rakesh Punia, Vivek K. Singh, Mahavir Bishnoi, Anoop Singh, Raju Ram Choudhary, Shiwani Mandhania

**Affiliations:** 1Department of Genetics and Plant Breeding, Oilseed Section, CCS Haryana Agricultural University, Hisar, Haryana 125004, India; ramavtar0706@gmail.com (R.A.); punia.rakesh98@gmail.com (R.P.); vks.slay@gmail.com (V.K.S.); mahaveer.bishnoi9@gmail.com (M.B.); rajuramchoudhary33@gmail.com (R.R.C.); 2Biochemistry Laboratory, Department of Genetics and Plant Breeding, Cotton Section, CCS Haryana Agricultural University, Hisar, Haryana 125004, India; 3Department of Biochemistry, College of Basic Sciences and Humanities, CCS Haryana Agricultural University, Hisar, Haryana 125004, India; ajaydrdo@rediffmail.com; 4Department of Botany, Maharshi Dayanand University, Rohtak, Haryana 124001, India; anoopnehra93@gmail.com

**Keywords:** *Sclerotinia sclerotiorum*, Indian mustard, lesion length, resistance, susceptible, antioxidant system, genotypes

## Abstract

Productivity of Indian mustard, an important oilseed crop of India, is affected by several pathogens. Among them, the hemibiotroph *Sclerotinia sclerotiorum*, which causes sclerotinia rot disease, is the most devastating fungal pathogen causing up to 90% yield losses. The availability of host resistance is the only efficient approach to control and understand the host–pathogen interaction. Therefore, the present investigation was carried out using six Indian mustard genotypes with contrasting behavior towards sclerotinia rot to study the antioxidant resistance mechanism against *S. sclerotiorum*. The plants at post-flowering stage were inoculated with five-day-old pure culture of *S. sclerotiorum* using artificial stem inoculation method. Disease evaluation revealed significant genotypic differences for mean lesion length among the tested genotypes, where genotype DRMR 2035 was found highly resistant, while genotypes RH 1569 and RH 1633 were found highly susceptible. The resistant genotypes had more phenolics and higher activities of peroxidase, catalase and polyphenol oxidase which provide them more efficient and strong antioxidant systems as compared with susceptible genotypes. Studies of antioxidative mechanisms validate the results of disease responses.

## 1. Introduction

Oilseed crops play a crucial role in the agricultural-based economy of India [1]. Among the nine major oilseed crops widely cultivated in India, rapeseed-mustard occupies the foremost position because of its greater sustainability under varied agro-ecological situations. India is ranked 3rd in world after China and Canada both in acreage (19.3%) and production (11.3%) of brassica oilseeds. Despite this, India meets ~57% of its interior comestible oil requirements through import from other countries [2]. Productivity of brassica oilseed is mainly hampered by various fungal diseases *viz.* white rust, Alternaria blight, sclerotinia rot, and downy mildew [3]. Among these, sclerotinia rot caused by *Sclerotinia sclerotiorum* (Lib.) de Bary, a cosmopolitan soilborne fungal pathogen, is the utmost devastating disease in the present climate [4]. This pathogen does not have any specific host and causes disease in >600 plant species including important oilseed crops *viz*. soybean, groundnut, sunflower, and brassica oilseeds including Indian mustard (*Brassica juncea* (L.) Czern & Coss) [5,6]. It causes up to 90% yield losses in Indian mustard besides affecting oil quality [7]. *S. sclerotiorum* shows bimodal infection in their hosts by carpogenic and myceliogenic means (Figure 1). The sclerotia overwinter inside the soil and either germinate myceliogenically to form mycelia which infect basal stems (soilborne infection) or germinate carpogenically to form apothecia to infect any part via appressoria formation (air borne infection) [8]. Sclerotinia rot symptoms, particularly those due to carpogenic infection, are visible after flowering stage. It causes infection in plant parts including stem with typical symptoms of initially soft and white-grayish lesions which then extend throughout the stem and lead to plant collapse [6]. *S. sclerotiorum* has a typical hemibiotrophic lifestyle [9] and shows a triphasic model for infection on host plants. This infection process includes the (i) opportunistic–saprophytic phase in which the pathogen makes contact and penetrates the host tissue, (ii) pathogenic phase in which it releases chief virulence factor (oxalic acid) and cell wall-degrading enzymes like pectinases which cause necrosis in host tissue, and (iii) saprophytic phase in which a pathogen takes nutrients from the host necrotic tissue for completing its life cycle and develops sclerotia which completely collapse the infected tissue [10]. Stems are most affected by this devastating disease and damage to stems is directly related to major field losses. The fungal infection generally leads to the development of lesions on stems which consistently extend longitudinally throughout the stem as the disease progresses. Resistant host restricts the lesion extension on stems over time with the help of callose development, lignification, and cambium formation at the edge of lesion for delaying growth of a pathogen [11,12]. Therefore, stem lesion length, rate of lesion development over time, and Area Under Disease Progression Curve (AUDPC) are very good parameters for assessing resistance and disease progression over time.

Metabolism of the host plant also gets modulated post-pathogen attack which leads to elevated levels of reactive oxygen species (ROS) *viz.* hydrogen peroxide (H_2_O_2_), superoxide radicals (O_2_^−^), hydroxyl radicals (OH^−^), and singlet oxygen (O^−^) near the pathogen infection site. These molecules are essential for plants to combat against phytopathogens [13].

However, production of ROS above its threshold level is deleterious for both the pathogen and host cells and causes oxidative stress [14]. Plants have very complex antioxidant defense mechanisms and systems to circumvent the oxidative stress [15,16]. They comprise antioxidative and defense enzymes as well as some non-enzymatic components. This system maintains cellular homeostasis, helps in exploiting its favorable role to plants against pathogenic infection and is very useful biochemical indicator for disease resistance [17,18,19,20,21]. The host plant illustrates a unique defense response during a different phase of hemibiotrophs infection. This response supported by host-hemibiotrophs interaction suggests that high ROS accumulation in host during initial phase of infection is inhibitory to pathogen while it is beneficial for pathogen during the later phase [22]. Therefore, timing and strength of defense response activation determine host plant resistance level against hemibiotrophs. However, understanding of the host-pathogen interaction and resistance mechanism involved in Indian mustard against *S. sclerotiorum* is still very scanty and needs further elucidation. Therefore, present study was designed and executed to evaluate the antioxidative response and resistance behavior at early, mid, and later stages of *S. sclerotiorum* infection in contrasting Indian mustard genotypes.

## 2. Results

### 2.1. Disease Assessment

Inoculated plants showed characteristics symptoms of sclerotinia rot as white-greyish lesions on stems while uninoculated plants were healthy with no symptoms. Statistical analysis showed significant genotypic differences for mean stem lesion length at 6, 12, and 18 days after inoculation (DAI) (Table 1). Based on mean lesion length at 18 DAI, genotype DRMR 2035 showed highly resistant response with <2.5 cm mean lesion length while the genotypes *viz*. RH 1222-28 and EC 597328 showed resistant response with mean lesion length between 2.5 to 5.0 cm. However, majority of the plants of resistant genotypes (DRMR 2035 and RH 1222-28) were asymptomatic at 6 DAI and showed slow resistance phenomenon. Among susceptible genotypes, RH 1566 was found susceptible with mean lesion length between 7.5 and 10.0 cm, while the other two genotypes *viz*. RH 1569 and RH 1633 exhibited highly susceptible response with >10.0 cm mean lesion length. AUDPC varied from 25.26 to 113.37 with lowest (25.26) in DRMR 2035 and highest (113.37) in RH 1569. Rate of increase per day in lesion length varied from 0.07 to 0.87 cm day^−1^ with maximum (0.87 cm day^−1^) in RH 1566 at 6 DAI and minimum (0.07 cm day^−1^) in DRMR 2035 at 18 DAI (Table 2).

### 2.2. Biochemical Analysis

#### 2.2.1. Peroxidase (POX) Activity

Analysis of variance (ANOVA) revealed highly significant interactions for POX activity among variables *viz*. sampling times (ST), plant inoculation (PI), and genotypes with different responses to sclerotinia rot (G) (Table 3). Data presented in Table 4 show the highest POX activity (5.98 µmol min^−1^ mg^−1^ protein) in DRMR 2035 inoculated with *S. sclerotiorum* at 6 DAI (6DAI/I/R2) and lowest (2.19 µmol min^−1^ mg^−1^ protein) in RH 1566 at 18 DAI (18DAI/UI/S1). There was no significant difference for POX activity in uninoculated plants of all genotypes, but activity increased in all the genotypes after pathogen inoculation at all duration (6, 12, and 18 DAI). However, it was significantly higher in resistant genotypes as compared with susceptible genotypes at 6 DAI while significantly higher in susceptible genotypes at 18 DAI (Table 4). The increase in POX activity after inoculation was manifold higher in resistant genotypes at 6 and 12 DAI, whereas at 18 DAI it was higher in susceptible genotypes (Figure 2). Pearson correlation between percent increase in POX activity and mean lesion length (Table 5) showed significantly negative association (*p <* 0.05) at 6 DAI.

#### 2.2.2. Catalase (CAT) Activity

Resistant genotypes had higher CAT activity as compared with susceptible genotypes at both inoculation treatments and all sampling times, but it significantly decreased after pathogen inoculation in a genotype-specific manner (Table 4). Percent decrease in CAT activity after inoculation was significantly higher in susceptible genotypes than resistant genotypes, and became linearly pronounced with disease progression (Figure 3). Highest CAT activity (7.08 µmol min^−1^ mg^−1^ protein) was observed in uninoculated plants of DRMR 2035 at 6DAI/UI/G2 while the lowest (1.23 µmol min^−1^ mg^−1^ protein) was observed in RH 1633 at 18DAI/I/S3. Activity decreased at later stages in all genotypes and in both treatments (Table 4). Percent decrease in CAT activity after inoculation also showed significantly positive correlation (*p <* 0.05) with mean lesion length at 6, 12, and 18 DAI (Table 5).

#### 2.2.3. Polyphenol Oxidase (PPO) Activity

Analysis of variance (ANOVA) showed that all interactions except sampling times by genotypes with different response to sclerotinia rot (ST X G) were not significant but all the variables *viz*. sampling times (ST), plant inoculation (PI), and genotypes with contrasting behavior to sclerotinia rot (G) were highly significant for PPO activity (Table 3). Data shows that PPO activity significantly increased after inoculation (Table 4). The highest activity (1.90 µmol min^−1^ mg^−1^ protein) was observed in resistant genotype (DRMR 2035) at 6DAI/I/R2 and thereafter it decreased in all genotypes and treatments. Increase in PPO activity after inoculation was ~2.4- and ~2.0-fold higher in resistant genotypes as compared with susceptible genotypes at 6 and 12 DAI, respectively. The linear curve shows a decrease in PPO activity as the plants matured after inoculation (Figure 4) and also reflects significantly negative association (*p <* 0.05) with mean lesion length at 6 DAI (Table 5).

#### 2.2.4. Total Soluble Phenolics (TSP)

Total soluble phenolics in resistant and susceptible genotypes significantly varied with respect to plant inoculation (Table 3). Resistant genotypes had significantly higher TSP as compared with susceptible genotypes in both the treatments along with all sampling times, and it increased with age of the plant (Table 4). Maximum TSP content (2.06 CE mg g^−1^ DW) was observed in resistant genotype (RH 1222-28) in treatment 6DAI/I/R1, while minimum (0.63 mg CE g^−1^ DW) in susceptible genotype (RH 1633) in treatment 6DAI/UI/S3 (Table 4). The overall increase in TSP after inoculation was higher in resistant genotypes as compared with susceptible genotypes, but it accumulated by ~2.4-fold in resistant genotypes as compared with susceptible genotypes at 6 DAI. Figure 5 depicts a declined per cent increase in TSP after 6 DAI.

#### 2.2.5. Total Soluble Sugar (TSS)

Analysis of variance (ANOVA) shows that interactions of sampling times and plant inoculation by genotypes with contrasting behavior to sclerotinia rot (ST X G; PI X G) along with individual variables like sampling times (ST), plant inoculation (PI) and genotypes with contrasting behavior to sclerotinia rot (G) were significant (Table 3). There was no significant difference between resistant and susceptible genotypes for TSS in uninoculated plants but there was a significant reduction in TSS in susceptible genotypes as compared with resistant genotypes after inoculation at all sampling stages (Table 4). Maximum TSS (49.74 mg GE g^−1^ DW) was found in resistant genotype EC 597328 in treatment 6DAI/I/R2 while minimum (12.43 mg g^−1^ DW) was recorded in susceptible genotypes RH 1569 at 18 DAI (18DA/I/S2). TSS declined 6 DAI in all genotypes, with more decrease in susceptible genotypes and highest difference was observed at 18 DAI (Table 4). Linear curve shows that percent decrease in TSS was more pronounced with post-inoculation passage of time (Figure 6) and it exhibited significantly positive correlation (*p <* 0.05) with mean lesion length at 6, 12, and 18 DAI (Table 5).

## 3. Discussion

Disease resistance and susceptibility are governed by genetics of host plants and fungal pathogens and depend on the exchange of multifaceted signals and the responses among them [24]. One of the major genotypic differences between disease resistant and susceptible plants is quick identification of infection and immediate activation of defense mechanisms [25]. Therefore, in the present study, we have investigated the activities of antioxidant enzymes *viz*. POX, CAT, and PPO as well as contents of non-enzymatic metabolites *viz*. TSP and TSS in six Indian mustard genotypes with contrasting behavior to sclerotinia rot for unraveling the biochemical basis of *B. juncea*–*S. sclerotiorum* interaction and antioxidant defense mechanism in Indian mustard against *S. sclerotiorum*.

High relative humidity (>80%); maximum and minimum temperatures of ~25 and 5–12 °C, respectively; and high soil moisture represent ideal conditions for *S. sclerotiorum* infection and disease progression in rapeseed-mustard crops [26]. Similar weather conditions were favorable for pathogenic infection on host during current field study (Appendix A). Present investigation shows significant genotypic differences for mean lesion length at different DAI which revealed it a host genotype-dependent trait. Lesion length and size during initial appearance to subsequent expansion have been used by crop scientists to categorize genotypic resistance against pathogens for a long time [27]. Therefore, we categorized our genotypes based on mean lesion length at 18 DAI where genotype DRMR 2035 was found highly resistant (<2.5 cm mean lesion length) while genotypes *viz.* RH 1222-28 and EC 597328 (<5.0 cm mean lesion length) were found resistant. This study also validates the results of Sharma et al. [28], where they tested these genotypes using stem inoculation techniques and found resistant against sclerotinia rot. Rate of increase in lesion length (cm day^−1^) is often used to observe pathogen aggression against host. It significantly affects the shape of disease progress curve (AUDPC values) and severity of disease [27]. During the present study, comparatively lower pace of lesion length (cm day^−1^) was shown by resistant genotypes which reveals a slow infection rate and lesser aggression of *S. sclerotiorum* against resistant genotypes as compared with susceptible genotypes. These genotypes were also differentiated based on AUDPC values which are very useful to summarize disease progression over time and to determine quantitative resistant phenotypes. Lower AUDPC values (<50) represent slower disease progression over time with greater resistance whereas higher AUDPC values (>100) represent faster disease progression and higher susceptibility against pathogens. Our study also revealed an interesting fact that some plants of resistant genotypes *viz*. DRMR 2035 and RH 1222-28 also showed purple coloration on stems near inoculation site which often extended up to 10 cm (Figure 7). It has earlier also been reported in resistant genotypes of *Brassica carinata* against this disease [29]. This observed variation for purple coloration and lesion length within the genotypes might be due to partial resistant nature of host. The genetic system controlling partial resistance is as diverse as phenotypes and there is no single genetic and phenotypic model to explain it. It is believed to be a blend of plant architecture escape and functional resistance. Partial resistance mechanism against *S. sclerotiorum* has earlier been reported in many oilseeds crops *viz*. oilseed rape [30,31], soybean [32,33,34], and sunflower [35,36]. The appearance of purple coloration near the infection site in resistant genotypes might be due to deposition of anthocyanin and comparatively higher TSS content as recently explained by Liu et al. [37] in oilseed brassica. Anthocyanin is a potent secondary metabolite and robust non-enzymatic antioxidant in plants which not only directly inhibits a pathogen but also protects the plants from elevated ROS accumulation in response to biotic stresses [38,39]. Inhibitory effect of anthocyanin against *S. sclerotiorum* has been well documented in soybean [40] and oilseed rape [37]. In present study, purple coloration in resistant genotypes after inoculation might be responsible for delaying *S. sclerotiorum* infection by inhibiting mycelial expansion.

One of the earliest responses of plants against pathogenic exposure is production of massive ROS in infected tissue [19], and although it seems to be the most efficient approach by a resistant host against biotrophs [41], it creates a dilemma for host plants infected by hemibiotrophs. The elevated ROS is not only injurious to the biotrophy to necrotrophy transition phase of hemibiotrophs but also equally destructive to host cells near infection site. It further benefits the necrotrophic phase because elevated ROS kills host cells which in turn are often fit for survival of hemibiotrophs during necrotrophic phase [42]. To maintain the ROS homeostasis, resistant hosts must have proper, efficient and timely regulated antioxidant machinery to utilize the ROS in their own beneficial way against the hemibiotrophs. Therefore, strength and timing of enzymatic antioxidant components *viz.* POX, CAT, PPO, and non-enzymatic components *viz*. phenolics and soluble sugar are vital in plant-pathogen interaction, especially in the case of pathogen having a hemibiotrophic lifestyle.

Analysis of variance showed significant effects of genotypes (*p =* 0.0002), plant inoculation (*p <* 0.0001), sampling times (*p <* 0.0001) and their interactions (*p <* 0.0001) on POX activity indicating it a host/genotype-dependent pathogen-mediated trait. Effect of genotypes and pathogen on its change with time is possibly because of dual feeding nature (biotrophy to necrotrophy) of a pathogen. POX is a well-recognized class of pathogenesis related (PR) protein and a potent plant protective enzyme against fungal pathogens. It is quick responsive and induce immediately near the infection site to catalyze detoxification of elevated H_2_O_2_ to protect the adjacent living cells from oxidative damage. Additionally, POX also causes lignification and suberization of cell wall to create a physical barrier against invading pathogens to locally restrict them. It also oxidizes the phenolic compounds to generate antifungal intermediates like phytoalexins and iso-flavonoids [43,44]. We observed an elevation of POX activity after *S. sclerotiorum* inoculation both in resistant and susceptible genotypes, but significantly higher in resistant genotypes especially during early infection stage. Percent deviation in POX activity after inoculation in resistant genotypes was significantly negative correlated with mean lesion length indicating its role in delaying *S. sclerotiorum* mediated lesion development by reducing mycelium expansion rate. It might be due to delay of transition from biotrophy to necrotrophy phase. However, at 18 DAI, POX activity was significantly higher in susceptible genotypes as compared with resistant genotypes. Therefore, it can be postulated that susceptible genotypes also evolve strategy of POX elevation to cope up with a pathogen and only difference between resistance and susceptible mechanism is the temporal activation of defense responses. It can also be linked with hemibiotrophic nature of a pathogen. ROS elevation during early stages is inhibitory to transition from biotrophy to necrotrophy phase while at later stage beneficial to necrotrophic phase as they survive and get their food from dead cells during necrotrophy. Our results reveal a strong link of POX activity with resistance and corroborate with previous reports on many important crops including soybean [45,46], common bean [47], sunflower [48,49,50,51], and oilseed rape [52,53].

CAT is another key enzyme which protects living cells by detoxifying H_2_O_2_ into water and molecular oxygen near pathogenic infection site. Its activity significantly changed depending on genotypic variation (*p* < 0.0001), plant inoculation (*p* < 0.0001) and their interaction (*p* < 0.0001) indicating it a genotype-specific and pathogen-dependent trait. Hydrogen peroxide, the chief substrate of CAT, is known to inhibit biotrophic but favor necrotrophic pathogens during host-pathogen interaction. The higher CAT activity observed in resistant genotypes as compared with susceptible genotypes before inoculation stipulates their higher H_2_O_2_ detoxifying capability. After inoculation, the decrease in CAT activity in inoculated plants over uninoculated plants was more pronounced in susceptible genotypes (Figure 3). It is apparent that there is a lesser decrease in resistant genotypes as compared with susceptible genotypes and elevated H_2_O_2_ might have helped in delaying infection during biotrophic phase. Shetty et al. [54] also found that hemibiotrophic phytopathogens are highly sensitive to H_2_O_2_ just after inoculation. In the case of susceptible genotypes, comparatively higher decrease in CAT activity and elevated H_2_O_2_ level cause oxidative burst in living cells adjoining the infection site and finally cell death to benefit the necrotrophic phase of *S. sclerotiorum*. Previous reports also demonstrate the role of CAT in resistance against *S. sclerotiorum* in important crops *viz*. sunflower [49,50,51,55], oilseed rape [52] and non-heading Chinese cabbage [56].

Analysis of variance revealed significant effects of genotype (*p <* 0.0001), sampling times (*p <* 0.0001) and their interaction (*p =* 0.0077) on PPO activity and showed it a genotype and duration specific traits. PPO catalyzes oxidation of phenolics compounds to highly reactive antimicrobial compounds like o-quinones and also takes part in defense mechanism against phytopathogens through several ways. Direct toxicity of o-quinones against phytopathogens, alkylation of plant cellular proteins, cross-linking of quinones with proteins and other phenolic compounds ultimately form a physical barrier against pathogens [57,58]. PPO activity in present study was significantly higher in resistant genotypes as compared with susceptible ones at 6, 12, and 18 DAI (Figure 4). Observation reveals that its increased activity causes higher production of toxic products (o-quinones and H_2_O_2_) and induces cell wall lignification after *S. sclerotiorum* infection and therefore, provides greater resistance. Increase in PPO activity after inoculation showed a significant negative correlation with mean lesion length at 6 DAI which reveals its impact in delaying biotrophy to necrotrophy transition phase of a pathogen. It can be suggested that high PPO activity is involved in constitutive resistance mechanism against a pathogen which is in agreement with previous reports on various important crops including oilseed rape [53,59,60], non-heading Chinese cabbage [56], sunflower [48] and soybean [46,61].

In this study, we found that TSP is also a host genotype-specific and pathogen mediated trait (Table 3). Its accumulation was significantly higher in resistant genotypes as compared with susceptible genotypes in both uninoculated and inoculated conditions. These phenolic compounds provide resistance to hosts by neutralizing elevated ROS produced after a pathogenic attack. Being structural components of cell wall to provide rigidity (lignin and suberin), TSP retards the activity of cell wall degrading enzymes of fungal pathogens during pathogenesis [62,63,64,65,66,67,68,69]. Increase in TSP after inoculation was higher in resistant genotypes than susceptible genotypes especially at 6 and 12 DAI. It provides the plant enough time for adoption of secondary strategies like phytoalexins synthesis and lignin depositions which might further restrict the invasion of *S. sclerotiorum* and spread to uninfected cells. It can be predicted that TSP are key metabolites for genotypic resistance in Indian mustard. Relatively higher phenolics content associated with *S. sclerotiorum* resistance in sunflower and *Phytophthora infestans* resistance in potatoes have been reported [70,71,72,73].

Sugars, besides their typical role as carbon and energy source, also take part in many metabolic and signaling pathways in plants [74]. They act both as oxidant and antioxidant and have capability to maintain ROS homeostasis in cells under stressful conditions. Sugars not only participate in ROS production, but are also involved in NADPH-producing pathways like oxidative pentose phosphate pathway to detoxify them [20]. The present study shows insignificant genotypic difference for TSS in uninoculated plants, but it decreased after a pathogen infection which was higher in susceptible genotypes at all stages. Decrease in TSS content in inoculated plants over uninoculated plants was also significantly higher in susceptible genotypes at 6 DAI and continued to be higher as disease progressed. Decline in TSS content at inoculation site might be either due to their utilization by a pathogen or due to diminished photosynthetic rate. A comparatively higher TSS level under diseased conditions in resistant genotypes shows better sugar resistance mechanism as it stimulates flavonoids and PR proteins synthesis against a pathogen [75]. A significant positive correlation between percent deviation of TSS content and mean lesion length reveals that TSS variation significantly affects mean lesion length (disease amount) as high sugar reduction is associated with susceptibility (high mean lesion length) and vice versa. Similar findings against *S. sclerotiorum* in sunflower [50], *Colletotrichum higginsianum* in thale cress [76], *Phomopsis theae* in tea [77], *Macrophomina phaseolina* and *Fusarium moniliforme* in sorghum [78], and *Plasmopara viticola* in grapes [79] have previously been reported.

Biological processes are very complex and metabolic pathways are often interlinked together. Therefore, it is worthy to discuss all above parameters together because they are highly linked with each other to provide resistance against *S. sclerotiorum*. Cell wall reinforcement through lignification and suberization is a H_2_O_2_-dependent POX-mediated mechanism, and phenolics and sugars are involved at least partially in both. Besides, CAT is a chief competitor of POX for H_2_O_2_ scavenging and PPO is involved in phenolics oxidation. Therefore, we can suggest that above biochemical indices are inter-linked together and a combination of these mechanisms is possibly associated with resistance against *S. sclerotiorum* in Indian mustard.

## 4. Materials and Methods

### 4.1. Plant Materials

Six Indian mustard genotypes with contrasting response to sclerotinia rot were used in present study. Three resistant genotypes *viz*. RH 1222-28, DRMR 2035, and EC 597328 and three susceptible genotypes *viz.* RH 1566, RH 1569, and RH 1633 were obtained from Directorate of Rapeseed-Mustard Research, Bharatpur (Rajasthan) and Oilseeds Section, Department of Genetics and Plant Breeding, CCS Haryana Agricultural University, Hisar (Haryana), respectively [28]. All genotypes were raised in Randomized Complete Block Design (RCBD) during *Rabi* season of 2019–2020 with plot size of 4 rows of 5 m length in three replications having row to row spacing of 30 cm. Crops were raised at the research farm of Oilseed Section, Department of Genetics and Plant Breeding, CCS Haryana Agricultural University, Hisar.

### 4.2. Sclerotinia Sclerotiorum Inoculum Preparation and Disease Assessment

Sclerotia of *S. sclerotiorum* (Hisar isolate) were collected from sclerotinia rot-infected plants of Indian mustard, rigorously washed in tap water, and surface sanitized by immersing in 0.1% HgCl_2_ solution for 80–90 s. They were then rinsed 3–4 times with sterilized water to remove traces of disinfectant and aseptically transferred into Potato Dextrose Agar (PDA) containing Petri plates. Plates were incubated for seven days in BOD incubator at 25 ± 1 °C. PDA slants were used for maintenance of sclertoia’s mycelium pure culture. Five-day-old cultures of *S. sclerotiorum* were artificially inoculated on stems of ten randomly selected plants from middle of each row at post-flowering stage as per method followed by Li et al. [80]. Stem lesion length (cm) was recorded at 6, 12, and 18 days after inoculation (DAI) using a linear ruler. The genotypes were classified into five different groups based on mean lesion length (cm) on stems at 18 DAI as per scale suggested by Garg et al. [23]. The groups were highly resistant (mean lesion length <2.5), resistant (2.5–5.0), moderately resistant (5.0–7.5), susceptible (7.5–10.0), and highly susceptible (≥10.0) with their respective rating 0, 1, 2, 3, and 4, respectively.

### 4.3. Sample Collection and Biochemical Analysis

Samples were collected from stems immediately next to infection sites at 6, 12, and 18 DAI. The collected samples were immediately frozen in dry ice box and stored in liquid nitrogen until further biochemical studies.

### 4.4. Determination of Defense-Related Antioxidant Enzymatic Activities

Stem tissue (0.2 g) was homogenized in 3 mL of 0.1 M potassium phosphate buffer (pH 7.0) and centrifuged for 15 min at 10,000 rpm (4 °C). Supernatant was used for assaying peroxidase (POX), catalase (CAT), polyphenol oxidase (PPO), and total soluble protein [81]. Total soluble protein was assayed as per standard protocol of Lowry et al. [82] using bovine serum albumin (BSA) as standard. POX activity (EC 1.11.1.7) was assayed as per method of Maehly [83] with slight modifications. Reaction mixture (3.25 mL) contained 50 µL enzyme extract, 0.1 M potassium phosphate buffer (pH 7.0), 0.5 M H_2_O_2_, and 0.9 M guaiacol. Increase in absorbance was recorded at 470 nm after every 15 s interval for 2 min. Enzyme activity was calculated by employing molar extinction coefficient (26.6 mM^−1^ cm^−1^) and expressed as µmol of tetra guaiacol formed per min per mg protein. CAT activity (EC 1.11.1.6) was assayed as per method given by Aebi [84]. The reaction mixture (3.55 mL) contained 50 µL enzyme extract, 0.1 M potassium phosphate buffer (pH 7.0), and 0.1 M H_2_O_2_. Decrease in absorbance was recorded at 240 nm for 2 min after every 15s interval. Enzyme activity was calculated using molar extinction coefficient (39.4 mM^−1^ cm^−1^) and expressed as µmol of H_2_O_2_ oxidized per min per mg protein. PPO activity (EC 1.10.3.2) was calculated according to method described by Gauillard et al. [85] with minor modifications. Reaction mixture contained 100 µL enzyme extract, 0.1 M potassium phosphate buffer (pH 7.0) and 0.025 M catechol. Increase in absorbance was recorded at 410 nm for 2 min after every 15 s interval. Enzyme activity was estimated using molar extinction coefficient (2.9 mM^−1^ cm^−1^) and expressed as µmol of quinone produced per min per mg protein.

### 4.5. Quantification of Total Soluble Phenolics (TSP) and Total Soluble Sugar (TSS)

Stem tissue (0.2 g) was extracted with 5 mL of 80% methanol (*v/v*) at 80 °C for one hour and centrifuged at 10,000 rpm for 10 min. Procedure was repeated twice. Supernatants were pooled, concentrated to 1 mL using vacuum evaporator and used for quantification of TSP and TSS. TSP was estimated following the method of Bray and Thorpe [86] using catechol as standard and expressed as mg catechol equivalent per gram tissue on dry weight basis (mg CE g^−1^ DW). TSS was estimated following the method of Dubois et al. [87] and expressed as mg glucose equivalent per gram tissue on dry weight basis (mg GE g^−1^ DW).

### 4.6. Statistical Analysis

Mean lesion length (cm) was measured by taking sum of each lesion length divided by number. Data was subjected to analysis of variance (ANOVA). For analysis of biochemical parameters, experiment was conducted under factorial design (3 × 2 × 6) consisting of three sampling times (ST) (6, 12, and 18 DAI), two inoculation treatments (PI) [uninoculated (UI) and inoculated plants (I)] and six Indian mustard genotypes with contrasting behavior to sclerotinia rot (G) (three resistant (R1, R2, and R3) and three susceptible genotypes (S1, S2, and S3)). The Duncan Multiple Range Test (DMRT) was used for multiple comparisons of means. Additionally, Pearson product-moment correlation analysis was performed to assess correlation between mean lesion length and percent deviation for biochemical parameters in inoculated plants over uninoculated plants. All data were analyzed using SAS version 9.4 Statistical Package (SAS Institute Inc., Cary, NC, USA). AUDPC was calculated by using mean lesion length at 6, 12, and 18 DAI. Percent deviation for biochemical parameters in inoculated plants over uninoculated plants was calculated using the formula given below.
(1)$$\mathrm{Percent}\;\mathrm{Deviation}=\frac{\mathrm{Mean}\;\mathrm{values}\;\mathrm{in}\;\mathrm{inoculated}\;\mathrm{plants}\;-\mathrm{mean}\;\mathrm{values}\;\mathrm{in}\;\mathrm{uninoculated}\;\mathrm{plants}}{\mathrm{Mean}\;\mathrm{values}\;\mathrm{in}\;\mathrm{uninoculated}\;\mathrm{plants}}\times 100$$

## 5. Conclusions

The findings from this study reveal that resistant genotypes have comparatively quicker, timely regulated, and strong antioxidant systems. They could restrict disease progression by higher activities of antioxidant enzymes and accumulating pigments after *S. sclerotiorum* infection. Higher TSP and highly induced POX activity especially during early infection and lesser depletion in TSS might be responsible for sclerotinia rot resistance in Indian mustard. This study improves upon the existing knowledge on antioxidant defense mechanisms; however, the transcriptomic profiling will be further line of research to confirm it.

## Figures and Tables

**Figure 1 pathogens-09-00892-f001:**
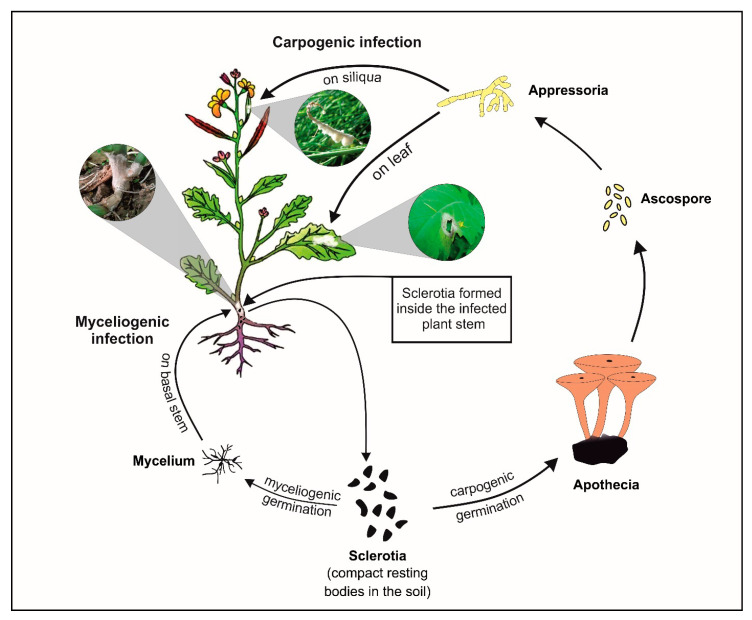
Lifecycle of *S. sclerotiorum* on Indian mustard.

**Figure 2 pathogens-09-00892-f002:**
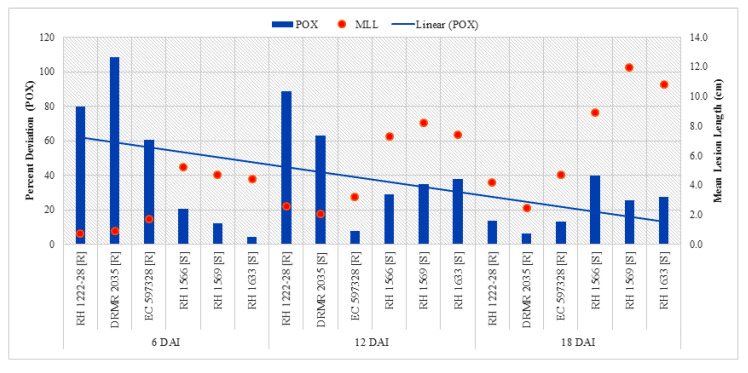
Comparison of percent increase in POX activity after *S. sclerotiorum* inoculation and mean lesion length (MLL) among different Indian mustard genotypes at 6, 12, and 18 DAI. (R—Resistant, S—Susceptible).

**Figure 3 pathogens-09-00892-f003:**
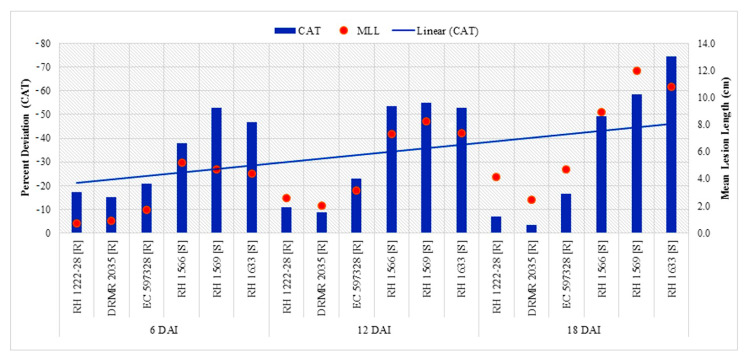
Comparison of percent decrease in CAT activity after *S. sclerotiorum* inoculation and mean lesion length (MLL) among different Indian mustard genotypes at 6, 12 and 18 DAI. R—Resistant, S—Susceptible.

**Figure 4 pathogens-09-00892-f004:**
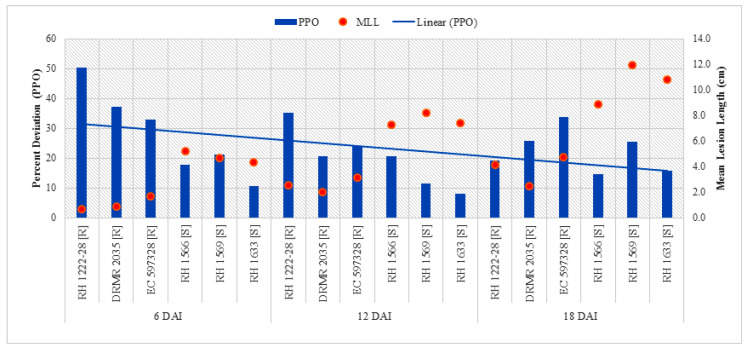
Comparison of percent increase in PPO activity after *S. sclerotiorum* inoculation and mean lesion length (MLL) among different Indian mustard genotypes at 6, 12, and 18 DAI. R—Resistant, S—Susceptible.

**Figure 5 pathogens-09-00892-f005:**
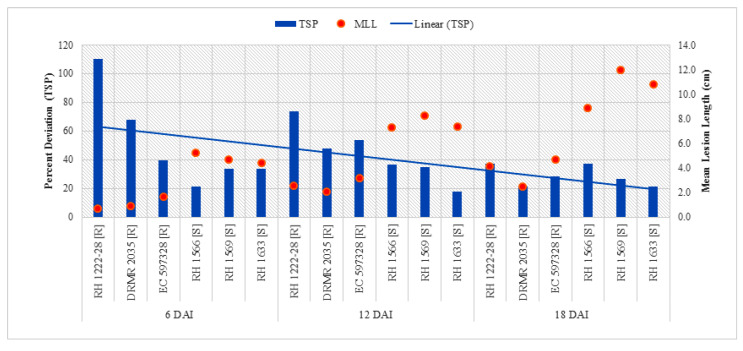
Comparison of percent increase in TSP after *S. sclerotiorum* inoculation and mean lesion length (MLL) among different Indian mustard genotypes at 6, 12, and 18 DAI. R—Resistant, S—Susceptible.

**Figure 6 pathogens-09-00892-f006:**
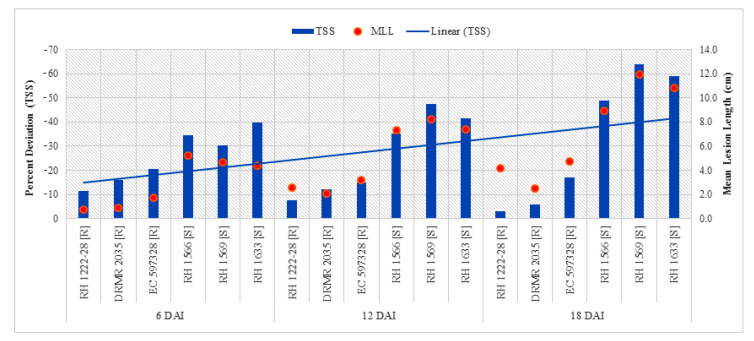
Comparison of percent decrease in TSS after *S. sclerotiorum* inoculation and mean lesion length (MLL) among different Indian mustard genotypes at 6, 12, and 18 DAI. R—Resistant, S—Susceptible.

**Figure 7 pathogens-09-00892-f007:**
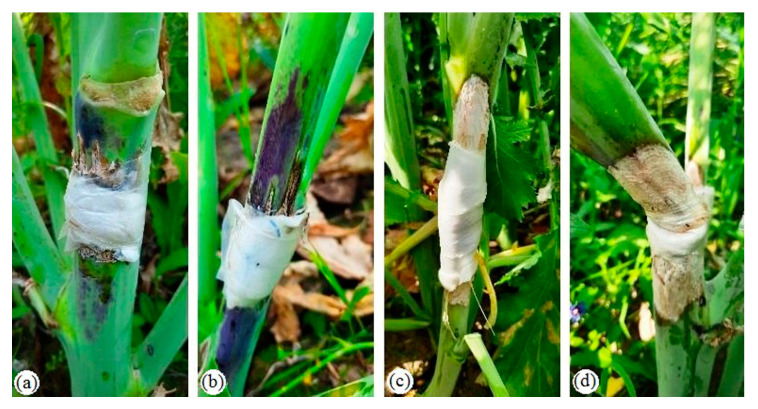
Resistance response of different Indian mustard genotypes against *Sclerotinia sclerotiorum*. (**a**) Extreme resistance shown by genotype RH 1222-28 and (**b**) DRMR 2035 with purple coloration extending on both sides of the inoculation site. (**c**) Typical susceptible response with lesion length >7.5cm shown by genotype RH 1566. (**d**) Stem breakage and high degree of susceptibility in genotype RH 1569 (plant collapsed).

**Table 1 pathogens-09-00892-t001:** Analysis of variance for mean lesion length on stems due to *S. sclerotiorum* inoculation.

Source of Variation	df	*p* Value
Lesion length (6 DAI)	5	<0.0001
Lesion length (12 DAI)	5	<0.0001
Lesion length (18 DAI)	5	<0.0001

DAI-Days After Inoculation, df-degree of freedom.

**Table 2 pathogens-09-00892-t002:** Disease progression characterization parameters for different Indian mustard genotypes at 6, 12, and 18 days after inoculation (DAI).

Genotype	Mean Lesion Length (cm)			Rate of Increase in Lesion Length (cm day^-1^)			AUDPC *	Disease Response **
	6 DAI	12 DAI	18 DAI	6 DAI	12 DAI	18 DAI		
RH 1222-28	0.72 ^d^	2.58 ^b^	4.16 ^c^	0.12	0.31	0.26	32.28	Resistant
DRMR 2035	0.91 ^d^	2.06 ^b^	2.48 ^c^	0.15	0.19	0.07	25.26	Highly Resistant
EC 597328	1.70 ^c^	3.19 ^b^	4.73 ^c^	0.28	0.25	0.26	43.53	Resistant
RH 1566	5.22 ^a^	7.30 ^a^	8.91 ^b^	0.87	0.35	0.27	101.85	Susceptible
RH 1569	4.68 ^b^	8.24 ^a^	11.95 ^a^	0.76	0.59	0.62	113.37	Highly Susceptible
RH 1633	4.39 ^b^	7.40 ^a^	10.81 ^ab^	0.73	0.50	0.57	103.17	Highly Susceptible

Treatment means in the same columns with different letters differ significantly (*p <* 0.05). * AUDPC—Area Under Disease Progress Curve, ** Disease response was measured on the basis of mean lesion length at 18 DAI as per scale given by Garg et al. [23].

**Table 3 pathogens-09-00892-t003:** Analysis of variance of the effect of sampling times (ST), plant inoculation (PI), and Indian mustard genotypes with different response to sclerotinia rot (G) and their interactions on the peroxidase activity (POX), catalase activity (CAT), polyphenol oxidase activity (PPO), total soluble phenolics (TSP), and total soluble sugar (TSS) content.

		*p* Values				
Sources of Variation	df	POX	CAT	PPO	TSP	TSS
ST	2	<0.0001	<0.0001	<0.0001	0.0283	<0.0001
PI	1	<0.0001	<0.0001	<0.0001	<0.0001	<0.0001
G	5	0.0002	<0.0001	<0.0001	<0.0001	<0.0001
ST X PI	2	0.0002	NS	NS	NS	NS
ST X G	10	<0.0001	NS	0.0077	NS	0.0120
PI X G	5	0.0003	<0.0001	NS	0.0001	<0.0001
ST X PI X G	10	<0.0001	NS	NS	NS	NS

NS—Nonsignificant, df—degree of freedom.

**Table 4 pathogens-09-00892-t004:** Mean comparison for interaction of sampling time (ST), plant inoculation, and Indian mustard genotypes (G) with contrasting behavior to sclerotinia rot on the peroxidase activity (POX), catalase activity (CAT), polyphenol oxidase activity (PPO), total soluble phenolics (TSP), and total soluble sugar (TSS).

DAI/PI/G	POX µmol min ^−1^ mg ^−1^ Protein	CAT µmol min ^−1^ mg ^−1^ Protein	PPO µmol min ^−1^ mg ^−1^ Protein	TSP mg g ^−1^ DW	TSS mg g ^−1^ DW
6DAI/UI/R1	2.96 ^e–g^	6.60 ^ab^	1.03 ^a−c^	0.98 ^b−d^	46.37 ^c^
6DAI/UI/R2	2.87 ^e−g^	7.08 ^a^	1.38 ^a−c^	1.00 ^b−d^	48.30 ^b^
6DAI/UI/R3	3.21 ^d−g^	6.22 ^a−c^	1.21 ^a−c^	0.94 ^b−d^	49.74 ^a^
6DAI/UI/S1	3.02 ^d−g^	5.98 ^b−d^	0.79 ^bc^	0.79 ^cd^	48.60 ^b^
6DAI/UI/S2	2.86 ^e−g^	6.05 ^b−d^	0.69 ^bc^	0.71 ^cd^	46.64 ^c^
6DAI/UI/S3	2.98 ^e−g^	5.84 ^b−f^	0.59 ^bc^	0.63 ^d^	47.82 ^b^
6DAI/I/R1	5.32 ^ab^	5.45 ^c−h^	1.56 ^ab^	2.06 ^a^	41.07 ^f−h^
6DAI/I/R2	5.98 ^a^	6.01 ^b−d^	1.90 ^a^	1.68 ^a−d^	40.62 ^gh^
6DAI/I/R3	5.16 ^ab^	4.93 ^e−i^	1.60 ^ab^	1.31 ^a−d^	39.51 ^i^
6DAI/I/S1	3.64 ^c−e^	3.71 ^k−m^	0.93 ^a−c^	0.96 ^b−d^	31.81 ^p^
6DAI/I/S2	3.21 ^d−g^	2.86 ^mn^	0.83 ^bc^	0.95 ^b−d^	32.50 ^op^
6DAI/I/S3	3.11 ^d−g^	3.11 ^l−n^	0.66 ^bc^	0.84 ^b−d^	28.73 ^q^
12DAI/UI/R1	2.63 ^e−g^	5.56 ^c−g^	0.94 ^a−c^	1.09 ^a−d^	41.34 ^fg^
12DAI/UI/R2	2.75 ^e−g^	5.87 ^b−e^	1.27 ^a−c^	1.18 ^a−d^	40.33 ^hi^
12DAI/UI/R3	2.98 ^e−g^	5.35 ^c−h^	0.92 ^a−c^	1.03 ^b−d^	44.08 ^d^
12DAI/UI/S1	2.54 ^fg^	5.13 ^d−h^	0.80 ^bc^	0.91 ^b−d^	42.50 ^e^
12DAI/UI/S2	2.70 ^e−g^	5.48 ^c−h^	0.63 ^bc^	0.78 ^cd^	40.62 ^gh^
12DAI/UI/S3	2.89 ^e−g^	4.93 ^e−i^	0.66 ^bc^	0.75 ^cd^	41.92 ^ef^
12DAI/I/R1	4.97 ^b^	4.96 ^e−i^	1.26 ^a−c^	1.89 ^ab^	38.16 ^j^
12DAI/I/R2	4.49 ^bc^	5.35 ^c−h^	1.54 ^ab^	1.74 ^a−c^	35.41 ^l^
12DAI/I/R3	3.21 ^d-g^	4.12 ^i−k^	1.14 ^a−c^	1.58 ^a−d^	37.45 ^j^
12DAI/I/S1	3.28 ^d−f^	2.39 ^no^	0.96 ^a−c^	1.24 ^a−d^	27.52 ^r^
12DAI/I/S2	3.64 ^c−e^	2.48 ^no^	0.70 ^bc^	1.05 ^a−d^	21.42 ^t^
12DAI/I/S3	3.99 ^cd^	2.33 ^no^	0.72 ^bc^	0.88 ^b−d^	24.46 ^s^
18DAI/UI/R1	2.51 ^fg^	4.86 ^f−j^	0.78 ^bc^	1.17 ^a−d^	34.35 ^m^
18DAI/UI/R2	2.39 ^fg^	5.31 ^c−h^	0.96 ^a−c^	1.23 ^a−d^	36.61 ^k^
18DAI/UI/R3	2.43 ^fg^	4.67 ^g−k^	0.67 ^bc^	1.06 ^a−d^	40.78 ^gh^
18DAI/UI/S1	2.19 ^g^	4.53 ^h−k^	0.59 ^bc^	0.98 ^b−d^	35.33 ^l^
18DAI/UI/S2	2.51 ^fg^	4.12 ^i−k^	0.46 ^c^	0.90 ^b−d^	34.43 ^m^
18DAI/UI/S3	2.30 ^fg^	4.81 ^g−j^	0.55 ^bc^	0.84 ^b−d^	35.32 ^l^
18DAI/I/R1	2.85 ^e−g^	4.51 ^h−k^	0.93 ^a−c^	1.60 ^a−d^	33.28 ^no^
18DAI/I/R2	2.53 ^fg^	5.12 ^d−h^	1.21 ^a−c^	1.49 ^a−d^	34.43 ^m^
18DAI/I/R3	2.75 ^e−g^	3.89 ^j−l^	0.89 ^a−c^	1.36 ^a−d^	33.87 ^mn^
18DAI/I/S1	3.06 ^d−g^	2.29 ^no^	0.67 ^bc^	1.34 ^a−d^	18.11 ^u^
18DAI/I/S2	3.16 ^d−g^	1.71 ^op^	0.58 ^bc^	1.14 ^a−d^	12.43 ^w^
18DAI/I/S3	2.94 ^e−g^	1.23 ^p^	0.64 ^bc^	1.02 ^b−d^	14.48 ^v^

Treatment means in the same columns with different letters differ significantly (*p <* 0.05). The design consisted of three sampling times (ST) (6, 12, and 18 days after inoculation (DAI)), two inoculation treatment (PI) (uninoculated plants (UI) and inoculated plants (I)), and six Indian mustard genotypes with contrasting behavior to sclerotinia rot (G) (3 Resistant (R1, R2, and R3) and 3 susceptible (S1, S2, and S3)) (R1-RH 1222-28, R2-DRMR 2035, R3-EC 597328, S1-RH 1566, S2-RH 1569, S3-RH 1633).

**Table 5 pathogens-09-00892-t005:** The Pearson’s product-moment correlation matrix between mean lesion length (cm) and percent deviation (PD) for peroxidase (POX), catalase (CAT), polyphenol oxidase (PPO) activities, total soluble phenolics (TSP), and total soluble sugar (TSS) in inoculated plants over uninoculated plants at different sampling times *viz.* 6, 12, and 18 DAI.

Per Cent Deviation (PD)	Mean Lesion Length (cm)		
	6 DAI	12 DAI	18 DAI
POX	−0.943 *	−0.407 ^NS^	0.801 ^NS^
CAT	0.999 *	0.976 *	0.965 *
PPO	−0.862 *	−0.742 ^NS^	−0.436 ^NS^
TSP	−0.697 ^NS^	−0.768 ^NS^	−0.045 ^NS^
TSS	0.887 *	0.972 *	0.976 *

* Significant linear relationship (*p <* 0.05) between PD in inoculated plants over uninoculated plants for enzymes (POX, CAT, PPO) activities and metabolites (TSP, TSS) contents with sclerotinia rot resistance. NS—Nonsignificant. Negative values indicate negative correlation of PD with mean lesion length (cm) and vice versa. Higher PD in POX, PPO, and TSP is directly associated with resistance (lesser mean lesion length) while positive values indicate positive correlation of PD with mean lesion length (cm) and vice versa. Lower PD in CAT and TSS is directly associated with sclerotinia rot resistance (lesser mean lesion length).

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
