# Peer review of "Genotype-Specific Antioxidant Responses and Assessment of Resistance Against Sclerotinia sclerotiorum Causing Sclerotinia Rot in Indian Mustard"

_pathogens, 2020, doi:10.3390/pathogens9110892_

Round 1

Reviewer 1 Report

This paper describes the association of several enzymatic and non-enzymatic plant-factors with the resistance to S. sclerotiorum in Indian mustard.

The conceptualization and the methodology are adequate, and the information is potentially interesting and could be considered for publication. However, a correct evaluation of the scientific content of this manuscript is very complicated. This difficulty is due to the presence in the text of multiple writing and grammatical errors that make it difficult to understand the text, to the use of extremely long sentences, to the unnecessary repetition of a same idea in a same paragraph (specially in the discussion section),  etc.

I kindly suggest that the manuscript be deeply revised with the advice of a native English speaker.

Author Response

I thank the learned reviewer for critically evaluating our work. We have thoroughly revised the manuscript where multiple writings and grammatical errors have been removed to make it easily understandable. The extremely long sentences also have been simplified for better understanding. The help of a native English speaker has been friendly sought.

Reviewer 2 Report

This study shows the relationship between an antioxidative activity and a host resistant availability against Sclerotinia sclerotiorum in Indian mustard species.

In this study, the activities of oxidative stress-related enzymes, POX, CAT, and PPO, have been determined. However, it is considered that only CAT could be an antioxidative enzyme. Furthermore, the activities of that enzyme were higher in the susceptible genotypes than resistant ones. Moreover, the activities of oxidative enzymes, POX and PPO, were stronger in resistant genotypes. The reviewer wondered why those enzymes showed such a controvertible behavior.

Moreover, regarding non-enzymatic antioxidant, the authors should have determined glutathione rather than TSS. It was so difficult to understand why a sugar could scavenge reactive oxide species.

Author Response

I thank the learned reviewer for critically evaluating our work. We have thoroughly revised the manuscript keeping in view your valuable suggestions. We do agree that availability of TSS plays a minor role in resistance but this is an upcoming area of research. Some of the available information are place below in this context: 

The protective effects of soluble sugars against oxidative stress have been mostly attributed to signalling effects, triggering the production of specific ROS scavengers [Couee et al. 2006; Ramel et al., 2009]. However, it was recently proposed that soluble sugars, especially when they are present at higher concentrations, might act as ROS scavengers themselves [Van den Ende and Valluru, 2009]. Endogenous sugar availability can feed the oxidative pentose phosphate pathway [Couee et al. 2006, Debnam et al., 2004], creating reducing power for glutathione (GSH) production, contributing to H2O2 scavenging. 

The activity of CAT is higher in resistant genotypes as compared with susceptible genotypes as shown in Table 4. The scale of Y-axis in Fig. 3 is negative.

Your suggestion on the role of glutathione in the said response is praiseworthy and will include in further experiments.

Round 2

Reviewer 1 Report

The manuscript has been significantly improved, but it still needs editing corrections, mainly in the discussion section.
In general, it is necessary to review the omission of the article "the" and of some verb in many sentences, and the correct use of commas, and semicolons in the longer sentences.
For example, on lines 276-285:
check the use of commas and semicolons on lines 276-277
at line 285, do you mean pathosystem or pathogen?
You should also add "the" before "case", and "a" before "pathogen".
Also check the text in lines 314 (yet?), 328-329 (designated it (as) a genotype and duration specific trait-?), 346 (do you mean "resistant" or "resistance"?).

Author Response

We thank the learned reviewer for again critically checking and improving our manuscript. We have carefully omitted the article 'the' in revised manuscript. We have taken utmost care to properly place the verb, comma and semicolons in the longer sentences. In lines 276-285, we have corrected the use of commas and semicolons in lines 276-277. In 285, the word pathosystem is replaced with pathogen. We have added "the" before "case", and "a" before "pathogen" and corrected the lines 314, 328-329 and 346. We hope the revised manuscript is fit for consideration now.

Reviewer 2 Report

The reviewer appreciates the response and some efforts which has been done by the authors to enhance the quality of the work.

Author Response

We thank the learned reviewer for improving our manuscript.